# OpenReview forum: "Evaluating Robustness of Reward Models for Mathematical Reasoning"
_ICLR.cc/2025/Conference — Submitted to ICLR 2025_

### Official Review · Reviewer_TTuB · 2024-10-31

**Soundness:** 2
**Presentation:** 3
**Contribution:** 2
**Rating:** 5
**Confidence:** 3

**Summary:**

The paper addresses a specific limitation of RewardBench, a widely used benchmark for reward model evaluation, in its assessment of mathematical reasoning capabilities. To this end, the authors introduce RewardMATH, a new benchmark that employs one-to-many comparisons of chosen and rejected responses to mathematical questions to enhance evaluation robustness. Experiments show that RewardMATH correlates well with policy performance and is more effective at identifying potential reward overoptimization and the reliability of reward signals.

**Strengths:**

1. The paper identifies a particular limitation of RewardBench, a popular benchmark for evaluating reward models, in assessing mathematical reasoning and introduces a new benchmark that addresses this issue by including one-to-many comparison data.
2. The authors provide extensive experiments and analyses across different reward model types, including both proprietary and open models, and assess various performance metrics.

**Weaknesses:**

1. The benchmark comparison primarily involves RewardBench, which is designed to evaluate reward models more holistically across various domains. However, is the comparison in terms of mathematical reasoning appropriate, given that RewardMATH is specifically designed for this purpose? If RewardBench is indeed the most comprehensive eval set even for mathematical reasoning tasks prior to RewardMATH, it would be helpful to clarify this point.
2. The benchmark appears to lack cases where reward models must distinguish between correct solutions of varying quality, such as those missing reasoning steps. It is also unclear whether 500 samples is sufficient to cover diverse mathematical reasoning tasks.
3. Tables 1 and 2 report performance comparisons of various LLMs on RewardBench and RewardMATH. The results seem to merely suggest that the two benchmarks differ significantly. Can we conclude from these results that "high scores on RewardBench do not guarantee **robustness** in reward models"?

**Questions:**

Q. How much of the relatively poor results for RewardBench is due to the noisy annotations inherited from PRM800K, as mentioned in Sec. 3.1? In other words, could simply fixing these annotations significantly change the comparison?

---

> ### Author Response · Authors · 2024-11-17
> **Response to Reviewer TTuB [1/2]**
>
> Dear Reviewer TTuB,
>
> We appreciate your comments and feedback on our work. We will address the questions raised by the reviewer below.
> ___
> ### **W1.**
> > ... However, is the comparison in terms of mathematical reasoning appropriate, given that RewardMATH is specifically designed for this purpose? If RewardBench is indeed the most comprehensive eval set even for mathematical reasoning tasks prior to RewardMATH, it would be helpful to clarify this point.
>
> Thank you for highlighting the points that required clarification. While reward models are used in a wide range of studies, there have been relatively few attempts to analyze or interpret them. So, RewardBench was the only available benchmark for reward models, and no benchmark existed that focused on specific domains, including mathematics. Consequently, the research community primarily relied on RewardBench to assess reward models. However, we argue that top-ranked models on the math subset of RewardBench may be vulnerable to reward hacking, and its one-to-one comparisons are unreliable. To address these limitations, we start with a focus on the analysis of reward models for mathematical reasoning tasks, as it provides a clear definition of human preference, without the need for deliberation on true preferences. We then propose guidelines for designing and structuring benchmarks tailored to reward models and validate them through comprehensive experiments.
>
> ___
>
> ### **W2.**
>
> > The benchmark appears to lack cases where reward models must distinguish between correct solutions of varying quality, such as those missing reasoning steps.
>
> As you mentioned, many-to-many comparisons are indeed preferable to one-to-many. However, as we outlined in Section 3.2 and the Limitation (Appendix A), gathering correct solutions requires significant human resources, so we focused on collecting a variety of rejected solutions. While many-to-many comparisons are undoubtedly the final goal, we emphasize that the most efficient next step, given the current reliance on RewardBench (i.e. one-to-one) as the primary benchmark, is to adopt a one-to-many comparison. We believe that this approach should advance consistently across all domains, not just in mathematics.
>
> > It is also unclear whether 500 samples is sufficient to cover diverse mathematical reasoning tasks.
>
> RewardBench was constructed from PRM800K, which is based on MATH500, so we used MATH500 as well to ensure a fair comparison. Furthermore, rather than constructing a comprehensive benchmark that covers all tasks, our goal is to propose the next step for reward model benchmark and to validate our design through a thorough analysis.

---

> ### Author Response · Authors · 2024-11-17
> **Response to Reviewer TTuB [2/2]**
>
> ___
> ### **W3.**
>
> > Tables 1 and 2 report performance comparisons of various LLMs on RewardBench and RewardMATH. The results seem to merely suggest that the two benchmarks differ significantly. Can we conclude from these results that "high scores on RewardBench do not guarantee **robustness** in reward models"?
>
> As mentioned earlier, both RewardBench and RewardMATH are constructed based on MATH500, and thus are not significantly different. Additionally, the rejected solutions in RewardBench were generated with an unaligned GPT-4, and GPT-family models were also used to produce the rejected solutions in RewardMATH. Consequently, the model rankings in RewardBench should not differ significantly from those in RewardMATH. However, for example, Oasst-rm-2.1-pythia-1.4b ranks among the top 3 in RewardBench but falls to second-to-last in RewardMATH. This discrepancy suggests that models achieving high scores in RewardBench may not actually be robust.
>
> ___
> ### **Q1.**
>
> > How much of the relatively poor results for RewardBench is due to the noisy annotations inherited from PRM800K, as mentioned in Sec. 3.1? In other words, could simply fixing these annotations significantly change the comparison?
>
> We find this question intriguing and consider it a valuable point. Instead of manually inspecting and correcting all misannotations, we can infer their impact from the table below. Similar to Figure 4 and Table 12, the table below presents the correlation between the results of the one-to-one benchmark, comparing RewardBench’s chosen solutions with each model’s rejected solutions in RewardMATH, and the performance of the optimized policy on the downstream tasks. We believe that even if the misannotations in PRM800K were fully corrected, the results would remain similar to those in this table, as other critical issues—such as susceptibility to reward hacking and the limitation of one-to-one comparisons—would still persist.
>
> Thanks to your insightful question, we believe we can make our work stronger, further supporting the importance of both reducing the possibility of reward hacking and employing multiple comparisons. From the table below, we can observe that significant representation gaps between chosen and rejected responses make reward models more susceptible to reward hacking. Moreover, regardless of which rejected solutions are used, the correlations remain substantially lower than those of one-to-many comparisons. This reconfirms the effectiveness of our design for reliable benchmark. We have now incorporated the table below into Table 12 of the updated PDF **([click to see the pdf \[link\]](https://openreview.net/pdf?id=0er6aOyXUD))**, with the detailed results provided in Appendix C.4.
>
>
> | chosen (RewardBench) vs.  |  | MetaMATH-Mistral-7B |  |  |  | WizardMATH-7B-v1.1 |  |
> | ----- | :---: | :---: | :---: | :---: | :---: | :---: |  :---: |
> | **rejected**  | **MATH500** | **Gaokao** | **SAT** |  | **MATH500** | **Gaokao** | **SAT** |
> | GPT-4o-2024-05-13 | \-0.174 | \-0.157 | \-0.245 |  | \-0.118 | \-0.041 | \-0.146 |
> | GPT-3.5-turbo-0125 | 0.135 | 0.126 | 0.124 |  | 0.264 | 0.256 | 0.247 |
> | Claude-3-sonnet-20240229 | 0.234 | 0.247 | 0.143 |  | 0.341 | 0.373 | 0.308 |
> | Meta-LLaMA3-70B | 0.058 | 0.066 | 0.096 |  | 0.165 | 0.185 | 0.236 |
> | Mixtral-8x7B | 0.193 | 0.187 | 0.124 |  | 0.291 | 0.317 | 0.236 |
> | Gemma-2-27b-it | 0.008 | 0.011 | \-0.022 |  | 0.099 | 0.124 | 0.082 |
> | DeepSeek-V2 | 0.292 | 0.313 | 0.286 |  | 0.434 | 0.446 | 0.463 |
> | Phi-3-medium | 0.074 | 0.071 | 0.069 |  | 0.209 | 0.196 | 0.187 |
> | Meta-LLama3-8B | 0.025 | 0.055 | 0.055 |  | 0.159 | 0.160 | 0.225 |
> | Qwen1.5-7B-chat | 0.316 | 0.330 | 0.259 |  | 0.434 | 0.446 | 0.396 |
> | Gemma-7b-it | 0.311 | 0.335 | 0.259 |  | 0.439 | 0.463 | 0.434 |
> | WizardMath-7B-v1.1 | 0.275 | 0.324 | 0.195 |  | 0.429 | 0.446 | 0.341 |
> | MetaMATH-Mistral-7B | 0.308 | 0.237 | 0.204 |  | 0.341 | 0.397 | 0.269 |
> | RewardMATH (random choice) | 0.162 | 0.170 | 0.107 |  | 0.264 | 0.287 | 0.247 |

---

> ### Author Response · Authors · 2024-11-21
> **Sincere Request for Review of Our Responses, New Experiments, and Revised Paper**
>
> Dear Reviewer TTuB,
>
> Thank you again for your time and effort to provide your insightful feedback on our paper.
>
> We have addressed your comments and incorporated the additional results into Table 12 of the updated PDF **([click to see the pdf \[link\]](https://openreview.net/pdf?id=0er6aOyXUD))**.
> If you have any remaining questions or require further clarification, we would be happy to address them before the time window closes.
>
> Thank you so much for your time and valuable feedback!
>
> Best regards,
>
> The Authors of Paper 10281

---

> ### Author Response · Authors · 2024-11-28
> **Sincere Request for Review of Our Responses and Revised Paper**
>
> Dear Reviewer TTuB,
>
> Thank you once again for your time and effort in providing insightful feedback on our paper. As you mentioned, we conduct additional experiments and incorporated the results into Table 12 of the updated PDF **([click to see the pdf \[link\]](https://openreview.net/pdf?id=0er6aOyXUD))** . Furthermore, we have provided responses to your additional questions.
>
> We hope that our responses sufficiently address your concerns. Should there be an opportunity for further discussion during the rebuttal period, we would be delighted to engage and provide any clarifications or further insights.
>
> Best regards,
>
> The Authors of Paper 10281

---

> ### Author Response · Authors · 2024-12-02
> **Gentle Reminder**
>
> Dear Reviewer TTuB,
>
> Thank you again for your thoughtful feedback on our paper. Your feedback has been invaluable in improving our work. As today is the final day for discussion, we would be delighted to provide any clarifications or further insights if needed. Please let us know if there are any remaining concerns we can address.
>
> Best regards,
>
> The Authors of Paper 10281

---

### Official Review · Reviewer_K97G · 2024-11-01

**Soundness:** 2
**Presentation:** 3
**Contribution:** 1
**Rating:** 5
**Confidence:** 4

**Summary:**

The paper introduces REWARDMATH, a benchmark for evaluating reward models in mathematical reasoning tasks, arguing that it provides a more reliable evaluation than the existing RewardBench by using one-to-many comparisons instead of pairwise comparisons. The authors validate their approach by showing correlation between benchmark performance and best-of-n sampling results.

**Strengths:**

The paper provides a good correlation study between the proposed benchmark and best of N downstream performance in datasets like MATH and Gaokao etc. The proposed one-to-many comparison seems to be on the right direction for better correlation with downstream performance.

**Weaknesses:**

I have the following major concerns:

1. Technical Contribution & Novelty: The primary contribution seems to be replacing pairwise comparisons in reward bench with best of N comparisons, which is an incremental modification rather than a substantial methodological advance. I do not think the change only is sufficient for a publication in top machine learning conference. The correlation between N-way comparison performance and best-of-N sampling results is somewhat expected and doesn't provide deep insights into reward model behavior.


2. Unclear Definition of Robustness: The paper uses "robustness" throughout but never provides a precise definition. The authors seem to equate robustness with correlation to best-of-n sampling results, but this is circular reasoning since the benchmark itself uses similar methodology. There's no clear framework for what properties a "robust" reward model should have beyond empirical correlation with certain metrics.


3. Limited Experimental Validation: The paper relies heavily on correlation with best-of-n sampling as validation, but doesn't explore other important aspects of reward model behavior. To make the paper deeper and broader, it would be great if the authors could try comparing the correlation of different real down-stream fine-tuning techniques like BoN SFT, DPO, PPO etc. and see whether how RewardBench and RewardMath correlate with downstream performance there. It would also be interesting to see if such observation extends to other domain like coding, and perhaps even open-ended generations without ground truth.

**Questions:**

Why do the authors choose 10 generations with 9 incorrect and 1 correct answer in the k-way comparisons? How does the choice of k and numbers of correct and incorrect answers affect the resulting correlations and reward overoptimization? The relationship between reward overoptimization and the proposed benchmark needs more rigorous analysis.

---

> ### Author Response · Authors · 2024-11-17
> **Response to Reviewer K97G [1/2]**
>
> Dear Reviewer K97G,
>
> First of all, we appreciate your constructive feedback on our work. We address the key issue raised by the reviewer in the comments below:
> ___
> ### **W1. Technical Contribution & Novelty**
>
> As you mentioned, if we aim to propose a new benchmark simply by changing the one-to-one comparisons in the existing benchmark (i.e. RewardBench) to one-to-many, we agree that this would be considered an incremental modification. However, what we aim to propose is not a new benchmark (i.e. RewardMATH), but rather insights into the future direction of reliable benchmark for reward models. So, we would like to provide a brief motivation and summary of our work.
>
> Despite many studies achieving sufficiently high scores on RewardBench, few have questioned the quality or evaluation methodology of RewardBench. Moreover, it is recently recognized that models performing well on RewardBench do not necessarily deliver strong results when applied in the RLHF system. Therefore, we first highlight the issues with RewardBench—such as poor quality, susceptibility to reward hacking, and unreliable one-to-one comparisons—and argue for a benchmark design that employs multiple comparisons for a more reliable evaluation, using RewardMATH as an example.
>
> The most notable aspect of RewardBench is the significant difference in step counts between chosen and rejected solutions, underscoring the importance of preventing reward hacking by ensuring that models do not exploit a preference for shorter-step solutions over solutions correctness (Figure 4). We also emphasize that evaluating models through one-to-one comparisons based on isolated solutions risks misjudging actual performance of the models, and we believe that many-to-many comparisons represent the ideal approach for reward model benchmark. However, since gathering a variety of correct solutions demands substantial human resources, we adopted a one-to-many comparison instead (Section 3.2 and Limitation). Using RewardMATH—a benchmark representing this one-to-many design—we conducted a thorough analysis through the lens of reward overoptimization, demonstrating that a design involving multiple comparisons results in a significantly more reliable benchmark.
>
> To summarize, we underscore the issues with RewardBench and demonstrate, through RewardMATH, that a benchmark design involving multiple comparisons is significantly more reliable. As a result, we argue that adopting a many-to-many comparison across all domains—not just mathematics—is the optimal path for constructing a reliable benchmark for reward models.
> ___
> ### **W2. Unclear Definition of Robustness**
>
> In Section 2.3, we discussed robustness, stating, ‘we argue that the robustness of a reward model should be evaluated based on how effectively it provides signals from which a policy can learn’. In other words, we consider a robust reward model to be one that is resistant to reward hacking, consistently assigning high scores to good answers and low scores to poor ones.

---

> ### Author Response · Authors · 2024-11-17
> **Response to Reviewer K97G [2/2]**
>
> ___
> ### **W3. Limited Experimental Validation**
>
> Thank you for the insightful comments. We understand the concern regarding the lack of PPO or DPO in our experiment setup and appreciate the opportunity to address this. Below, we would like to address any concerns and clarify our experimental setup:
>
> **Challenges of PPO in a Non-Synthetic Setup**
> Many previous studies have used the responses of an SFT model to train the same pretrained model as the reward model to achieve stable RLHF (PPO) training \[1-2\]. In particular, \[2\] highlights that initializing the reward model with the same pretrained model helps prevent information mismatches with the policy model, contributing to a consistent and accurate reward signal. Additionally, \[2\] and \[3\] suggest that as the policy model improves, the data distribution shifts, and if the reward model is not exposed to this new distribution, its accuracy may be limited.
> In our case, the reward models we evaluate are trained on different backbone (i.e. pretrained) models and are also different from the policy model, making stable PPO training challenging in a non-synthetic setup. Indeed, when we attempted training with several reward models, the training process was highly unstable. Due to these reasons, it was difficult to perform comprehensive PPO experiments with various reward models.
> We have now clarified the challenges of PPO in a non-synthetic setup in Appendix B.4 of the updated PDF **([click to see the pdf \[link\]](https://openreview.net/pdf?id=0er6aOyXUD))**.
>
> **Concern that BoN Sampling Involves Circular Reasoning**
> As the reviewer mentioned in weakness 2, the methodology for evaluating RewardMATH—specifically, identifying the correct solution from multiple options—seems similar to the approach used in best-on-$n$ sampling. To clarify, however, BoN sampling involves generating multiple responses from a single model (i.e. a policy) and selecting the one with the highest reward, whereas RewardMATH utilizes responses generated by a wide range of models. Thus, the two approaches do not employ the same methodology, and it does not constitute circular reasoning.
>
> **Correlation with Downstream Tasks Other than BoN**
> We understand the reviewer’s concern regarding the need to verify the effectiveness of RewardMATH beyond BoN sampling, through methods such as DPO or PPO. As previously mentioned, due to the instability of PPO experiments in our setup, we focused on conducting experiments where the reward model can effectively provide learning signals.
>
> * **Preference data for DPO constructed using the reward model:** We created a preference dataset for DPO by selecting a response with the highest reward as the “chosen” sample and a response with the lowest reward as the “rejected” sample.
>
> In our experiments, we used MetaMATH-Mistral-7B as the SFT model and selected a 32K subset of data from the MetaMATH dataset as the training dataset, considering the short discussion (rebuttal) period. We performed n=32 sampling with the SFT model and removed instances that were entirely correct or incorrect to reduce noise and better assess whether the reward model provides meaningful learning signals. Finally, we obtained rewards from each reward model for a final dataset of 13.5K responses and conducted training with DPO.
>
> The table below presents the correlation between the results of the optimized policies on MATH500 and the benchmark results. As a result, we reconfirm that the results on DPO also show a stronger correlation than RewardBench.
>
> |  | RewardBench | RewardMATH |
> | :---- | ----- | ----- |
> | DPO with reward model | 0.156 | 0.725 |
> | BoN | 0.187 | 0.902 |
>
> **References**
>
> \[1\] Ouyang, Long, et al. "Training language models to follow instructions with human feedback." Advances in neural information processing systems 35 (2022): 27730-27744.
>
> \[2\] Touvron, Hugo, et al. "Llama 2: Open foundation and fine-tuned chat models." arXiv preprint arXiv:2307.09288 (2023).
>
> \[3\] LeVine, Will, et al. "A Baseline Analysis of Reward Models' Ability To Accurately Analyze Foundation Models Under Distribution Shift." arXiv preprint arXiv:2311.14743 (2023).
>
> ___
> ### **Q1.**
> > Why do the authors choose 10 generations with 9 incorrect and 1 correct answer in the k-way comparisons? How does the choice of k and numbers of correct and incorrect answers affect the resulting correlations and reward overoptimization?
>
> As noted in Limitation (Appendix A), we consider finding the optimal $k$ to be beyond the scope of our study, given that our goal is not to propose a well-crafted new benchmark. As the number of solutions ($k$) increases, both the inference cost and the reliability of the results rise. Therefore, identifying the optimal trade-off point is crucial, and future work on the construction of benchmark should take this into account.

---

> ### Author Response · Authors · 2024-11-21
> **Sincere Request for Review of Our Responses, New Experiments, and Revised Paper**
>
> Dear Reviewer K97G,
>
> Thank you again for your time and effort to provide your insightful feedback on our paper.
>
> We have addressed your comments and clarified the challenges of PPO in a non-synthetic setup in Appendix B.4 of the updated draft **([click to see the pdf \[link\]](https://openreview.net/pdf?id=0er6aOyXUD))**.
> If you have any remaining questions or require further clarification, we would be happy to address them before the time window closes.
>
> Thank you so much for your time and valuable feedback!
>
> Best regards,
>
> The Authors of Paper 10281

---

> > ### Comment · Reviewer_K97G · 2024-11-26
> >
> > Thank you for the response. I appreciate the efforts the authors have made to verify the effectiveness of the reward bench in downstream RLHF performance with new DPO experiments, and the investigation of the new benchmark scores. I have updated my score accordingly.
> >
> > However, I'm not fully convinced that the contribution is enough as a ICLR conference paper, due to the following reason:
> >
> > 1. The core technical contribution is still the change from [1 to 1] to [1 to many], and as authors conjected, perhaps [many to many].
> > 2. The main focus is only math performance, while reward bench does focus on a wide range of chat capabilities including hard, reasoning, safety etc.

---

> ### Author Response · Authors · 2024-11-26
>
> Dear Reviewer K97G,
>
> Thank you for your response and for reconsidering the score. In addition to the clarification provided in the general response and W1, we would like to provide further clarification regarding the points you raised.
> ___
> > ...core technical contribution is ... the change from [1 to 1] to [1 to many] ...
>
> We understand that our contribution might be seen as merely a change from one-to-one to one-to-many comparisons. However, we would like to emphasize that this change is not our only contribution. We have described our contributions below:
>
> - **We identify the limitations of the existing benchmark.** We experimentally verified that the scores on RewardBench show a weak correlation with the performance of policies on downstream tasks, using BoN sampling and DPO. Moreover, we figured out several limitations of RewardBench, including poor quality, vulnerability to reward hacking, and unreliable one-to-one comparisons (i.e. evaluation based on single, isolated cases).
> - **We propose a better design for a reliable benchmark.** RewardBench has a significant difference in step counts between chosen and rejected solutions. So, we argue that the benchmark for reward models should be careful of the huge difference between chosen and rejected solutions to prevent reward hacking. Moreover, since evaluating a reward model using a single pair of solutions is highly prone to misjudgement, we underscore the need for multiple comparisons.
> - **We thoroughly validate the design through correlation with downstream tasks and through the lens of reward overoptimization.** Due to the limitations of RewardBench, its scores show a weak correlation with performance on downstream tasks. However, our design, which mitigates reward hacking factors (i.e. the huge difference between chosen and rejected responses) and incorporates multiple comparisons, achieves a strong correlation. Furthermore, Reward overoptimization is a critical challenge in RLHF, where models may overfit to specific reward signals, leading to degraded generalization. So, it is important to effectively estimate the degree of reward overoptimization via the scores on the benchmark. Through the lens of reward overoptimization, we show that the limitations of RewardBench raises critical issues that compromise its role as a reliable benchmark.
>
> (As we mentioned in the paper, we refer to the math subset of RewardBench simply as RewardBench. So, there may be domains within RewardBench that do not suffer from issues like poor quality or the huge difference between chosen and rejected responses.)
>
> To summarize, we describe our key insights below:
> 1. **In benchmarks for reward models, a significant difference between the chosen and rejected responses shows low correlation with downstream tasks due to the potential for reward hacking.**
> 2. **One-to-one comparisons may yield inaccurate results depending on the preference pairs, which in turn results in low correlation with downstream tasks.**
> 3. **A benchmark employing multiple comparisons can effectively capture reward overoptimization, indicating its ability to assess the robustness of reward models.**
> ___
> > The main focus is only math performance, while reward bench does focus on a wide range of chat capabilities including hard, reasoning, safety etc.
>
> We believe it would be unfair to compare this work with RewardBench in terms of the scope of the domains they address. The authors of RewardBench propose a new benchmark, which consists of a preference set to assess reward models across various domains. But it lacks a comprehensive investigation into the reliability of its results, including vulnerabilities to reward hacking, insufficient analysis of correlations with downstream tasks, and an inability to effectively estimate reward overoptimization.
>
> In contrast, although our focus was exclusively on mathematics, we emphasized delivering profound insights into the considerations for benchmarks used to assess reward models. We also believe that our insights—validation against reward hacking and reward overoptimization—can be applied to a variety of domains in the future.
>
> &nbsp;
>
> Why mathematics?
> As we mentioned in the general response, the reason we chose mathematics is as follows:
>
> - **One of the tasks where reward models are most extensively used is mathematics reasoning.** To enhance mathematical reasoning capabilities, reward models are widely utilized during training (e.g. PPO) and at inference-time through reward-based techniques such as BoN sampling or Monte Carlo Tree Search (MCTS).
> - **Mathematical reasoning includes a clear human preference.** In mathematical reasoning, human preference can be easily defined as correctness, allowing us to focus effectively on the analysis without the need to deliberate over true preferences.
>
> &nbsp;
>
> Thank you once again for your time and thoughtful feedback, and for engaging with our submission.
>
> &nbsp;
>
> Best regards,
>
> The Authors of Paper 10281

---

> ### Author Response · Authors · 2024-12-02
> **Gentle Reminder**
>
> Dear Reviewer K97G,
>
> Thank you again for your thoughtful feedback on our paper. Your feedback has been invaluable in improving our work. As today is the final day for discussion, we would be delighted to provide any clarifications or further insights if needed. Please let us know if there are any remaining concerns we can address.
>
> Best regards,
>
> The Authors of Paper 10281

---

### Official Review · Reviewer_BzTe · 2024-11-02

**Soundness:** 2
**Presentation:** 2
**Contribution:** 3
**Rating:** 6
**Confidence:** 3

**Summary:**

The paper proposes a new benchmark, REWARDMATH, to improve the robustness evaluation of reward models in mathematical reasoning tasks. It highlights limitations in the existing RewardBench benchmark, which relies on single comparisons between chosen and rejected solutions, potentially leading to reward hacking and misjudgments of model robustness. REWARDMATH addresses this by using one-to-many comparisons with multiple incorrect solutions to better capture robustness and reduce the risk of reward over-optimization. Experimental results indicate that scores on REWARDMATH strongly correlate with policy optimization outcomes and provide a more reliable measure of reward model robustness compared to RewardBench. This benchmark aims to enhance the development and reliability of RLHF systems, with the findings underscoring the potential of REWARDMATH to serve as a trustworthy evaluation tool in this domain​.

**Strengths:**

1. **Thoroughness**: The paper presents detailed implementations, including training hyperparameters and experimental protocols. This ensures that other researchers can accurately reproduce the experiments and validate the findings.
2. **Relevance**: This work addresses a critical gap in the field by focusing on reward model evaluation, a crucial area of research that has significant implications for the development of more reliable AI systems.
3. **Motivation**: The paper presents a compelling critique of the existing Reward Bench evaluation metric, establishing a strong foundation for their work. The authors make a persuasive case for developing benchmarks that minimize over-optimization risks, backing their arguments with experimental evidence. This dual focus on improving metrics while addressing practical concerns demonstrates clear motivation for the research.

**Weaknesses:**

1. **Clarity**: The paper is generally well written, however, it has some clarity issues, especially in section 5, which is hard to follow. Clarification questions are asked in the question section, marked with [Clarification]. The authors should address those questions.
2. **Benchmark Biases**: The paper has several biases, raising concerns on the claimed robustness and reliability. Examples and comments below:

> Line 206: Hence, we first convert the human-annotated solutions from MATH500 into step-by-step machine-generated solutions. We prompt GPT-4, using 4 carefully crafted exemplars for each math subject as part of the prompt.

All correct solutions in the benchmark are generated via GPT-4, raising concerns regarding biases towards GPT-series models. Even though the authors manually inspect the solutions, the solutions were still mainly generated using GPT-4. Notably, the authors observe LLM judges from the GPT-4 Series to perform significantly higher than other models (Line 286), which is likely due to this oversight (since it is known LLM judges tend to bias their own response, eg. GPT-4 family judge favors responses from GPT-4 family). The authors should use a diverse set of LLMs to curate the correct solutions to avoid potential biases.

> Line 805: Secondly, we instruct GPT-4-0125-preview to select a specific step from the correct solution, transform it into an erroneous step, and then prompt again to continue generating the solutions from the erroneous step.

Similar to the previous point, employing GPT-4-0125-preview as editor to insert errors into other LLMs’ answers may introduce biases. Additional validation is needed to ensure the benchmark does not exhibit any bias towards GPT family models.

> Line 402: We assume Internlm2-7B-reward, which performs well on both RewardBench and REWARDMATH, as the gold RM.

The use of Internlm2-7b-reward as the gold standard lacks sufficient justification and raises several concerns about experimental validity. The author relies primarily on performance metrics from RewardBench and REWARDMATH, but this approach is problematic for multiple reasons. First, the authors themselves criticized RewardBench for containing incorrect ground truth data and failing to adequately assess reward models. Second, using REWARDMATH as a benchmark is circular since it's the very dataset being studied. High scores on these benchmarks alone don't necessarily indicate that a reward model can reliably approximate human preferences. To establish Internlm2-7b-reward as a legitimate gold standard, the author should conduct additional validation studies specifically demonstrating its ability to align with human judgment on mathematical tasks.

> Line 426: We find that proxy reward models trained on smaller datasets reach peak rewards at lower KL divergences, indicating faster over-optimization.

The author assumes KL divergence adequately captures optimization degree without proper justification. KL may not account for other important aspects of policy change. Further study will strengthen the experimental results.

3. **Comprehensiveness**: This paper's scope is notably narrow, focusing solely on evaluating reward models' performance on mathematical problems. While it attempts to address limitations in a small subset of the Reward Bench dataset, its improvements remain constrained. The study primarily concentrates on reward over-optimization, overlooking other potential vulnerabilities in reward model benchmarking. Additionally, the benchmark's methodology of comparing one correct solution against multiple incorrect ones limits its thoroughness. Furthermore, the author's assumption that MATH500 adequately represents mathematical reasoning tasks may be oversimplified. These limitations collectively suggest a need for a more comprehensive approach to reward model evaluation.

**Questions:**

1. [Clarification] Are the prompts used to evaluate the LLM judge on REWARDMATH the same as the prompt used to evaluate the LLM judge on the Reward Bench? Different prompting strategy (eg. difference system prompt) raises concerns regarding fair comparison between the two benchmarks.
2. [Clarification] What is MATH500? The author did not mention the details behind this dataset which they used for their benchmark. Were there any steps taken to ensure the dataset is not contaminated with the models being evaluated? If the dataset is used during training on any of the evaluated RMs, the benchmark’s reliability will be undermined.
3. [Clarification] What was the motivation behind different parts of the Synthetic Data experiment? What was the reasoning behind using the MetaMATH dataset? Why was only 80K out of the 155K data points augmented from MATH used for training?
4. The authors did not mention how the incorrect solutions are ensured to be actually incorrect. Were there steps taken to validate the said incorrect solutions are indeed incorrect?

---

> ### Author Response · Authors · 2024-11-17
> **Response to Reviewer BzTe [1/4]**
>
> Dear Reviewer BzTe,
>
> We appreciate your comments and feedback on our work. We will address the clarification points you raised (i.e. weakness 1) at the end of our response.
> ___
> ### **W2. Benchmark Biases**
>
> **1\) Dataset construction**
>
> We understand the concern regarding potential bias introduced by using GPT-4 to convert human-written solutions into machine-generated step-by-step solutions for RewardMATH.
> However, we have already addressed this by conducting experiments of self-enhancement bias in LLM-as-Judge, with the detailed results provided in Appendix C.2. Figure 8 and Table 8 demonstrates that GPT-4o and other LLM judges exhibit a mild preference for their own rejected and correct solutions, but the bias is not significant. For example, when we used GPT-4o as LLM-as-Judge and performed a pairwise comparison (win or lose) between its own correct solution and the chosen solution from LLaMA3-70B, it selected only 48% of its own correct solution. Nonetheless, we agree on the importance of collecting diverse correct solutions and have noted this as a limitation in our paper (Appendix A). Furthermore, as shown in Figure 2b, the incorrect solutions modified by GPT-4 account for only 9% of the total incorrect solutions, which we believe minimizes any potential bias.
>
> **2\) Selecting Internlm2-7B as the gold RM and using KL Divergence to capture optimization degree**
> Prior studies on reward overoptimization often use a gold RM with more parameters than the proxy RM \[1, 2\] or even GPT-4 as the gold RM \[3, 4\]. However, conducting experiments with a larger model as the gold RM is challenging, as we need to assess a variety of reward models. Moreover, GPT-4 did not outperform other classifier-based reward models, making it unsuitable for gold RM. Therefore, we chose Internlm2-7B, which demonstrated high performance on both RewardBench and RewardMATH. We (and Reviewer BzTe) recognized that the previous studies’ approach of considering only gold reward may not be the optimal. To address this, we introduce oracle reward based on human preferences, which, in the case of mathematics, align with accuracy, as detailed in Section 5.2.1.
> Additionally, although expanding the optimization metrics could offer further insights, we believe KL divergence is a sufficient measure here, as it has been widely used in prior research \[1-4\] to observe reward overoptimization.
>
> **References**
>
> \[1\] Gao, Leo, John Schulman, and Jacob Hilton. "Scaling laws for reward model overoptimization." Proceedings of the 40th International Conference on Machine Learning. 2023\.
>
> \[2\] Coste, Thomas, et al. "Reward Model Ensembles Help Mitigate Overoptimization." The Twelfth International Conference on Learning Representations.
>
> \[3\] Rafailov, Rafael, et al. "Scaling laws for reward model overoptimization in direct alignment algorithms." arXiv preprint arXiv:2406.02900 (2024).
>
> \[4\] Rame, Alexandre, et al. "WARM: On the Benefits of Weight Averaged Reward Models." Forty-first International Conference on Machine Learning.

---

> ### Author Response · Authors · 2024-11-17
> **Response to Reviewer BzTe [2/4]**
>
> ___
> ### **W3. Comprehensiveness**
>
> > … scope is notably narrow, focusing solely on evaluating reward models' performance on mathematical problems …
>
> Thank you for highlighting these concerns. However, there seems to be a slight misunderstanding regarding our work, so we would like to clarify these points. First, we want to make it clear that our work does not aim to introduce a new, meticulously designed benchmark but rather to provide insights into the future direction for developing reliable benchmarks for reward models.
>
> Since the success of RLHF, research utilizing reward models has been steadily growing. In this context, we observed that reward models are increasingly applied during both training and inference-time for reasoning tasks, particularly to enhance mathematical reasoning through Process Reward Models (PRM) or similar approaches \[5-8\]. Despite their importance, there has been limited analysis of reward models themselves. Furthermore, mathematical reasoning tasks allow human preferences to be clearly defined (i.e. correctness), enabling more focused and in-depth analyses. For these reasons, we chose to analyze reward models specifically within the context of mathematical reasoning.
>
> At the beginning of our research, the math subset of RewardBench was only the benchmark available for evaluating reward models for mathematical reasoning tasks. However, as we used this benchmark, we identified several limitations: (1) a significant distribution gap between chosen and rejected responses, (2) unexpectedly high performance for certain reward models, and (3) the potential to fail in accurately evaluating a model’s actual capabilities due to the limitations of the one-to-one comparisons. This observation led us to consider how to design a reliable benchmark and what analytical perspectives could be used to validate it.
>
> To address these issues, we found that reducing distributional discrepancies help prevent the possibility of reward hacking, while the one-to-many comparisons provide a reliable result. Additionally, by analyzing through the lens of reward overoptimization, we confirmed why reward models that perform well on RewardBench (i.e. the math subset) may still lack robustness. We hope our findings offer valuable insights to the research community, contributing to the development of more reliable benchmarks for reward models in the future.
>
> &nbsp;
> > … primarily concentrates on reward over-optimization, overlooking other potential vulnerabilities ... Additionally, … comparing one correct solution against multiple incorrect ones limits its thoroughness.
>
> Apologies, we find ourselves unclear about the specifics of these comments. Our confusion stems from not understanding what other potential vulnerabilities we may have overlooked, as well as why comparing a single correct solution with multiple incorrect solutions would limit the thoroughness of our approach. Thus, we kindly request that you provide more detailed explanations. We eagerly anticipate further clarification or discussion on this matter.
>
> &nbsp;
> > the author's assumption that MATH500 adequately represents mathematical reasoning tasks may be oversimplified.
>
> Yes, we agree that the problems in MATH500 do not fully represent all mathematical reasoning tasks. However, as previously mentioned, defining the scope of mathematics lies beyond the focus of our work, which aims to explore the future direction of reliable benchmark for reward models. Moreover, since RewardBench was constructed based on MATH500, we also utilize MATH500 as the foundation for RewardMATH to ensure clarity in our experiments and analyses.
>
> &nbsp;
>
> **References**
>
> \[5\] Lightman, Hunter, et al. "Let's Verify Step by Step." The Twelfth International Conference on Learning Representations.
>
> \[6\] Luo, Liangchen, et al. "Improve Mathematical Reasoning in Language Models by Automated Process Supervision." arXiv preprint arXiv:2406.06592 (2024).
>
> \[7\] Wang, Chaojie, et al. "Q\*: Improving multi-step reasoning for llms with deliberative planning." arXiv preprint arXiv:2406.14283 (2024).
>
> \[8\] Zhang, Dan, et al. "Rest-mcts\*: Llm self-training via process reward guided tree search." arXiv preprint arXiv:2406.03816 (2024).

---

> ### Author Response · Authors · 2024-11-17
> **Response to Reviewer BzTe [3/4]**
>
> ___
> ### **Q1.**
> > Are the prompts used to evaluate the LLM judge on REWARDMATH the same as the prompt used to evaluate the LLM judge on the Reward Bench? Different prompting strategy (eg. difference system prompt) raises concerns regarding fair comparison between the two benchmarks.
>
> RewardBench was evaluated using pairwise comparisons, whereas RewardMATH was assessed with both pairwise comparisons and direct assessment. The same prompt was applied for pairwise comparisons in both benchmarks (Figure 17), while a separate prompt was used for direct assessment in RewardMATH (Figure 16), following \[9\]. As noted in the Appendix B.3, Prometheus-2 utilizes the prompts proposed by \[10\] applying criteria specific to reasoning tasks (Figure 18, 19).
>
> **References**
>
> \[9\] Zheng, Lianmin, et al. "Judging llm-as-a-judge with mt-bench and chatbot arena." Advances in Neural Information Processing Systems 36 (2023): 46595-46623.
>
> \[10\] Seungone Kim et al. Prometheus 2: An open source language model specialized in evaluating other language models. arXiv preprint arXiv:2405.01535, 2024\.
>
> ___
>
> ### **Q2.**
> > What is MATH500? Were there any steps taken to ensure the dataset is not contaminated with the models being evaluated? If the dataset is used during training on any of the evaluated RMs, the benchmark’s reliability will be undermined.
>
> We apologize for any confusion resulting from the initial lack of detail on MATH500. We have now provided a citation and comprehensive information in Section 3 and Appendix B of the revised PDF **([click to see the pdf \[link\]](https://openreview.net/pdf?id=0er6aOyXUD))**.
>
> MATH500 is drawn from the original MATH dataset \[11\], which consists of a 7.5K training set and a 5K test set. During the construction of PRM800K \[12\], the original 7.5K training set was insufficient for training a robust Process Reward Model (PRM) on step-by-step solution data. Therefore, 4.5K problems from the MATH test set were incorporated into the training set, leaving a final subset of 500 problems now identified as MATH500. Since the release of PRM800K, MATH500 has been widely adopted to prevent overlap between training and test sets.
> To clarify, MATH500 originates from the original MATH dataset and is not a new contribution of our work. Additionally, only PRM800K contains mis-annotated samples, while the MATH500 subset, directly sourced from the original MATH dataset, remains accurate and unaffected.
>
> **References**
>
> \[11\] Hendrycks, Dan, et al. "Measuring Mathematical Problem Solving With the MATH Dataset." Thirty-fifth Conference on Neural Information Processing Systems Datasets and Benchmarks Track
>
> \[12\] Lightman, Hunter, et al. "Let's Verify Step by Step." The Twelfth International Conference on Learning Representations.

---

> ### Author Response · Authors · 2024-11-17
> **Response to Reviewer BzTe [4/4]**
>
> ___
> ### **Q3.**
> > What was the motivation behind different parts of the Synthetic Data experiment?
>
> In synthetic setup, we examined how well performance on the benchmark estimates reward overoptimization within a controlled environment. Following \[13\], preference data was labeled by a gold RM serving as a substitute for human annotators, and this data was used to train proxy RMs with varying amounts of training data, ranging from 8K to 65K. Table 3 presents the scores of proxy RMs on RewardBench and RewardMATH, and Figure 5 illustrates how the gold reward changes for each proxy RM as KL divergence increases. In conclusion, this synthetic setup demonstrates, under controlled conditions with all other variables held constant, that scores on RewardMATH offer a more accurate estimate of reward overoptimization.
>
> **References**
>
> \[13\] Gao, Leo, John Schulman, and Jacob Hilton. "Scaling laws for reward model overoptimization." Proceedings of the 40th International Conference on Machine Learning. 2023\.
>
> &nbsp;
> > What was the reasoning behind using the MetaMATH dataset? Why was only 80K out of the 155K data points augmented from MATH used for training?
>
> We appreciate your interest in the details of our work. We agree that this information is important, thus we detailed it in Appendix B.4.
> We can not train the policy model solely on the MATH dataset to generate synthetic preference data, as the limited amount of MATH was not sufficient for the policy to learn effectively, preventing the generation of meaningful synthetic data. So, we used 80K of the 155K MetaMATH dataset to train the policy model, with the remaining 75K used to generate synthetic preference data that reflects the policy’s preferences. We generated 16 samples per problem using the policy model and, after excluding samples that could not form preference pairs, created a final synthetic preference dataset of 65K pairs.
>
> ___
> ### **Q4.**
> > Were there steps taken to validate the said incorrect solutions are indeed incorrect?
>
> Yes, we were also very mindful of the point you raised. Thus, we provide the details in Appendix B.1.
> First, we filtered out the solutions with correct answers and selected only those with incorrect answers. As shown in the solution examples in Figure 1a, the final step of each solution includes the reasoning that leads to the answer. So, an incorrect final answer generally indicates that the solution is incorrect. However, the final answer is parsed from the solution using \[14\], which can sometimes lead to misclassifying the correct solutions as incorrect due to parsing errors, leading to their unintended inclusion. Hence, we conducted a manual inspection of all chosen and rejected solutions.
>
> **References**
>
> \[14\] Boning Zhang et al. MARIO Eval: Evaluate Your Math LLM with your Math LLM--A mathematical dataset evaluation toolkit. arXiv preprint arXiv:2404.13925 (2024)

---

> ### Author Response · Authors · 2024-11-21
> **Sincere Request for Review of Our Responses, New Experiments, and Revised Paper**
>
> Dear Reviewer BzTe,
>
> Thank you again for your time and effort to provide your insightful feedback on our paper.
>
> We have addressed your comments and added comprehensive details about MATH500 in Section 3 and Appendix B of the updated draft **([click to see the pdf \[link\]](https://openreview.net/pdf?id=0er6aOyXUD))**.
> If you have any remaining questions or require further clarification, we would be happy to address them before the time window closes.
>
> Thank you so much for your time and valuable feedback!
>
> Best regards,
>
> The Authors of Paper 10281

---

> ### Comment · Reviewer_BzTe · 2024-11-22
> **Official Response from Reviewer**
>
> Thank the author for their response.
>
> I understand the paper's new direction towards providing insights into future directions for constructing reliable benchmarks for reward models. However, I believe that if this is indeed the authors' goal, then the benchmark built in the paper must be also sound.
>
> W1. What was the detail behind this experiment? It is unclear from the paper which dataset did the experiments of self-enhancement bias in LLM-as-Judge utilized. How was the correctness determined? Was it MATH? If so there is a distribution difference between the experiment's dataset versus the benchmark dataset.
>
> W2. Thanks for the explanation. Without access to true oracle reward, using a "gold" RM is the only feasible alternative for most researchers. Nonetheless, I am still skeptical about the justification of using Internlm2-7b-reward, and feel the author should conduct additional validation. However, this seem fine to me for now.
>
> W3. To answer the author's questions, I was referring to the following:
>
> First, evaluating RM does not limit to reward over-optimization. Other attributes fundamentally matters, such as alignment to human preference and being able to distinguish between correct answers.
>
> Second, the paper is still limited in scope. The author stated in the meta comment:
> > Mathematical reasoning includes a clear human preference. In mathematical reasoning, human preference can be easily defined as correctness, allowing us to focus effectively on the analysis without the need to deliberate over true preferences.
>
> While I agree with the authors correctness might be the primary attribute to human preference in the context of mathematics, it is not going be true for other domains, such as creative writing and social studies. Even among technical fields, correctness is not the only attribute for open-ended tasks, such as web development and UI/UX design. On these tasks, human preference no longer only consist of correctness, but also includes other complex attributes like helpfulness (in the case of UI/UX, the visual result of the code generated by the model matters). The paper is limited to mathematics, and also close-ended tasks. I remain skeptical whether the methods proposed in the paper can be truly adopted by someone who is trying to build reward model benchmark in domain outside close-ended mathematical problems. It is not clear to me how to approach building reward benchmarks for other domains after reading the methods in the paper.
>
> I agree with Reviewer 1Mfa that to demonstrate the method is indeed generalizable as author claim, applying the method in another domain, ideally more open-ended queries, will be necessary.
>
> Thanks for answering my questions, the authors' answers are very helpful.

---

> ### Author Response · Authors · 2024-11-23
>
> Thank you for your response!
>
> ### **\[W1\]**
>
> > What was the detail behind this experiment? It is unclear from the paper which dataset did the experiments of self-enhancement bias in LLM-as-Judge utilized. How was the correctness determined? Was it MATH? If so there is a distribution difference between the experiment's dataset versus the benchmark dataset.
>
> Thank you for your thoughtful question and we apologize for any confusion caused by the lack of detailed explanation regarding the experiments of self-enhancement bias in LLM-as-Judge. We conducted this experiments with two research questions to analyze self-enhancement bias in LLM-as-Judge:
>
> 1. **Does the model prefer its own incorrect answers over correct answers?** In this experiment, we compared the model’s own rejected (incorrect) solutions from RewardMATH with correct solutions from RewardMATH. Note that this experiment involves not only preference bias but also the model's judgment ability, which must be considered when interpreting the results.
> 2. **When given two correct solutions, which one does the model prefer?** For this experiment, we collected correct solutions from the LLM-as-a-Judge model itself across 100 problems in MATH500 where all LLMs we used generated correct solutions. We examined the model's preference between its own correct solutions and correct solutions generated by other models. Since both solutions are correct, we evaluated them under two settings: (1) when a tie is an available option (w/ tie), and (2) when a tie is not an available option (w/o tie), to analyze which solution the model prefers more strongly in each setting.
>
> For the first experiment, we used a subset of RewardMATH's dataset that aligns with the rejected and correct solutions relevant to the LLM-as-a-Judge. For the second experiment, we collected correct solutions directly from the respective models, which is different from RewardMATH. All problems in these experiments are part of the MATH500 dataset.
>
> Furthermore, we conducted an additional experiment to address concerns about potential bias in the GPT-4 judge, as the correct solutions in RewardMATH were modified into machine-generated solutions using GPT-4, not GPT-4o. Our previous experiments of self-enhancement bias were evaluated by the GPT-4o judge; therefore, to ensure a more thorough analysis, we further validate using the GPT-4 judge. Based on the second research question mentioned earlier, we analyze which solution is preferred: the chosen solution from RewardMATH or the correct solutions from each model. The table below demonstrates that GPT-4 does not exhibit a bias toward preferring its own solutions. The results indicate that there is no bias, as the solutions were not directly generated by GPT-4 but were instead modified versions of human solutions in MATH dataset, which demonstrates that the benchmark is free from potential bias and affirms the fairness of the experiments.
>
> We have also incorporated the table below into Table 8 of the updated draft **([click to see the pdf \[link\]](https://openreview.net/pdf?id=0er6aOyXUD))**.
>
> |  | GPT-4o | GPT-3.5-turbo | Llama3-70B | Llama3-8B | Claude-3-Sonnet | Gemma2-27B |
> | :---: | :---: | :---: | :---: | :---: | :---: | :---: |
> | w/o tie (self / others) | 30.09 / 69.91 | 33.03 / 66.97 | 33.63 / 66.37 | 27.35 / 72.65 |  24.35 / 75.65 | 29.2 / 70.8 |
> | w/ tie  (self / tie / others) | 35.79 / 18.95 / 45.26 | 37.89 / 14.74 / 47.37 | 40.0 / 18.95 / 41.05 | 33.68 / 23.16 / 43.16 | 29.47 / 21.05 / 49.47 | 34.74 / 18.95 / 46.32 |

---

> ### Author Response · Authors · 2024-11-23
>
> ___
> ### **\[W3\]**
>
> > ... evaluating RM does not limit to reward over-optimization. Other attributes fundamentally matters, such as alignment to human preference and being able to distinguish between correct answers.
>
> As you commented, we agree that the evaluation of a reward model should include its alignment with human preference as well as its ability to distinguish between correct answers. However, this work focuses on **the benchmark** for evaluating reward models. Fundamentally, a benchmark for reward models is designed to evaluate how effectively the reward model aligns with human preferences. But, the existing benchmark (i.e. RewardBench) have assessed how well a reward model aligns with human preferences by using a single pair of predefined human preference (a single chosen vs. a single rejected).
>
> Thus, this work proposes a multiple comparison design for a reliable benchmark to better capture human preferences and validates our design through the following two perspectives:
>
> 1. Correlation with BoN sampling and DPO (included in the updated draft)
> 2. Reward overoptimization
>
> The results of BoN sampling involve selecting the best sample from numerous responses, which includes the reward model’s ability to distinguish between correct answers. Additionally, the results of reward overoptimization experiments reflect how robustly the reward model provides useful signals for policy learning. Thus, we believe that we have taken into account the attributes you mentioned.
>
> &nbsp;
>
> > ... paper is still limited in scope …
>
> As you know, mathematical reasoning is not a simple task. Mathematical reasoning serves as a cornerstone for deeper reasoning abilities, and many studies leverage reward models to enhance this capability \[1-5\]. However, the analysis of reward models remains under-explored. While the existing benchmark for reward models, RewardBench, has shown a weak correlation with downstream tasks, our design exhibits a strong correlation. We believe this represents a valuable contribution to future research on reward models.
>
> Furthermore, we assume that preferences are well-defined for the target tasks. Previous work has simply assessed human preferences through an isolated case (i.e. one-to-one comparisons) without considering two critical aspects: (1) **how well the benchmark correlates with downstream tasks** and (2) **how effectively the benchmark reflects the robustness of the reward model**. Hence, the findings—that a benchmark for reward models should capture their correlations with downstream tasks and their robustness against reward overoptimization—can reasonably be considered applicable to other domains.
>
> Therefore, we believe our work can provide valuable insights not only for mathematical reasoning but also for the broader evaluation of reward models across diverse tasks. We hope this response addresses your concerns and highlights the contributions and potential impact of our work.
>
> Best regards,
>
> The Authors of Paper 10281
>
> &nbsp;
>
> **[Updates]** We provide additional clarification on **the key insights** of this work in the general response.
>
> &nbsp;
>
>
> **References**
>
> [1] Lightman, Hunter, et al. "Let's Verify Step by Step." The Twelfth International Conference on Learning Representations.
>
> [2] Luo, Liangchen, et al. "Improve Mathematical Reasoning in Language Models by Automated Process Supervision." arXiv preprint arXiv:2406.06592 (2024).
>
> [3] Wang, Peiyi, et al. "Math-shepherd: Verify and reinforce llms step-by-step without human annotations." Proceedings of the 62nd Annual Meeting of the Association for Computational Linguistics (Volume 1: Long Papers). 2024.
>
> [4] Setlur, Amrith, et al. "Rewarding Progress: Scaling Automated Process Verifiers for LLM Reasoning." arXiv preprint arXiv:2410.08146 (2024).
>
> [5] Wang, Chaojie, et al. "Q*: Improving multi-step reasoning for llms with deliberative planning." arXiv preprint arXiv:2406.14283 (2024).

---

> > ### Comment · Reviewer_BzTe · 2024-11-28
> > **Official Response from Reviewer**
> >
> > Thanks for the response.
> >
> > W1. Thank you for the information, could authors point the necessary experiments and clarifications regarding bias to the said experiments in main text of the paper. Furthermore, mentioning only employing more models other than GPT-4 for crafting answer might be more ideal in limitation section. Also the limitation section is not linked to the main text of the paper, which makes it hard to find.
> >
> > W3. Thanks for the explanation. I still believe human preference is an important attribute regardless. Could author also address this point in the limitation section?
> >
> > > Therefore, we believe our work can provide valuable insights not only for mathematical reasoning but also for the broader evaluation of reward models across diverse tasks. We hope this response addresses your concerns and highlights the contributions and potential impact of our work.
> >
> > Could the authors provide a detailed example of how their approach could be applied to a domain outside of mathematics, preferably a non-technical one? Specifically, it would be helpful to see:
> >
> > 1. A concrete example in another domain
> > 2. How to mitigate reward hacking in that context
> > 3. How to collect unbiased reject and prefer pairs for that domain
> >
> > A specific example would help readers better understand how to apply this meta-approach more broadly.
> >
> > Depending on the authors' response, I am ready to re-evaluate my score accordingly. However, I will not change my score at the current time. I will not ask the authors to revise their paper in the short period of time, but they should provide how they would revise their paper if they were accepted.

---

> ### Author Response · Authors · 2024-11-28
> **Official Comment by Authors**
>
> Dear Reviewer BzTe,
>
> Thank you for your suggestions that could further enhance our work. We believe your comments addressed potential concerns that readers might have, and we have updated the draft accordingly **([click to see the pdf \[link\]](https://openreview.net/pdf?id=0er6aOyXUD))**. The updates made to the draft are as follows:
>
> - **Section 4.2 & Appendix C.3**
>   In Section 4.2, we added a short explanation about the self-enhancement bias in LLM judges and further clarification in Appendix C.3.
> - **Section 5.3**
>   We added a footnote regarding the explanation of our key insights and application to other domains.
> - **Appendix A.1 (Limitation)**
>   We provide further explanation on the considerations that should be carefully considered when applying our key insights to other domains in the future, as well as self-enhancement bias and future directions.
> - **Appendix A.2**
>   We elaborate on the reasons for selecting the mathematics domain, summarization of our key insights, how these insights can be applied to other domains, and an example of their application in the chat domain.
> - **Appendix C.5**
>   Although it is not part of your suggestions, we additionally examine the correlations between the performance on the benchmark and scores of optimized policy using BoN sampling on GSM8K.
>
> Thank you once again for your time and thoughtful feedback, and for engaging with our submission.
>
> Best Regards,
>
> The Authors of Paper 10281

---

> ### Author Response · Authors · 2024-12-02
> **Gentle Reminder**
>
> Dear Reviewer BzTe,
>
> Thank you again for your thoughtful feedback on our paper. Your feedback has been invaluable in improving our work. As today is the final day for discussion, we would be delighted to provide any clarifications or further insights if needed. Please let us know if there are any remaining concerns we can address.
>
> Best regards,
>
> The Authors of Paper 10281

---

> > ### Comment · Reviewer_BzTe · 2024-12-02
> > **Official Comment by Reviewer BzTe**
> >
> > I thank the authors for making these changes for the revision, I think it helps readability, and I like the discussion in the appendix. I have raised my score by 1 point.

---

### Official Review · Reviewer_1Mfa · 2024-11-04

**Soundness:** 2
**Presentation:** 3
**Contribution:** 2
**Rating:** 5
**Confidence:** 4

**Summary:**

This paper proposes RewardMATH, a reward model evaluation benchmark focused on the math domain. This benchmark adopts a one-to-many comparison to evaluate reward models robustly. Specifically, it provides 10 responses for each prompt where only 1 response is the answer. The evaluated reward model is considered accurate only when the answer response is given the highest score among all 10 responses.
The authors also empirically provide sufficient evidence that the RewardMATH benchmark provides reliable estimates for reward models that give policies optimized using BoN are indeed robust in math benchmarks.

**Strengths:**

- The paper provides clear and sufficient empirical evidence that their RewardMATH benchmark is more reliable than the math subset of RewardBench [1]. The empirical results are also clear as LLM policy using BoN on high-scored reward models on RewardBench shows little to no correlation with the performance increase of Math benchmarks (r-square = 0-0.1), while RewardMATH shows a much stronger correlation
(r-square = 0.6-0.8) in Figure 3.
- The authors have evaluated diverse reward models on RewardMATH, including LLMs (generative reward models), classifier-based reward models, and process reward models.
- The paper considers the problem of over-optimization using a synthetic setup of gold RMs and proxy RMs.

[1] https://arxiv.org/abs/2403.13787

**Weaknesses:**

- The work would be more interesting if the authors showed any other domains (such as coding or text summarisation or maybe safety) reward model benchmark can be improved by the framework proposed here (by adopting multiple responses and using diverse LLMs to generate outputs). Any initial or limited experiments would be helpful.

- The lack of PPO (or DPO) usage for policy fine-tuning in experiments seems like a major weakness. The main contribution of this paper is using policy fine-tuning methods to verify if the RewardMATH benchmark scores correlate with the signals it provides during policy fine-tuning. I agree with this approach and am impressed by the number of experiments conducted to verify this using mainly Best-of-N sampling. However, Best-of-N sampling is an inference time method to generate better model outputs using reward models, whereas PPO (or possibly DPO) is the main fine-tuning method researchers use. Although Figure 5 does show a PPO experiment under a synthetic setup, the number of checkpoints or whether the dots follow the findings from Gao et al [2] is not clear to me. Without any solid PPO results, Best of N sampling seems not enough to verify the benchmark's capability of measuring the robustness of reward models. The work will be much more convincing if the authors show more PPO-trained policy evaluations. Or at least, it will be helpful if the author provides more context as to why PPO is hard to train in their non-synthetic setup. Also, I suspect high-scoring reward models on RewardMATH have the ability to find the best response from multiple responses, and Best-of-N adopts a very similar way as it picks the response with the highest reward, resulting in a high correlation of results. Whether this ability will generalize even on PPO setups is not clear to me at this point.

- Experiment results in Figure 6 compare diverse RMs on both RewardBench and RewardMATH benchmarks with gold or oracle rewards. It would be nice if the authors not only provided the numbers but also a statistical analysis (such as Kendell's tau) that measures the agreement between RewardMATH(or RewardBench) and oracle (or gold) reward scores in Figure 6.

[2] https://arxiv.org/abs/2210.10760

**Questions:**

- As the proxy reward model trained from synthetic data shows far from optimal performance in Table 3 (only around 13% for RewardMATH and 69% for RewardBench), can you consider using better proxy RMs? The increase in value from 12.68 to 13.51 is not very convincing to me that this is a strong trend.

- In Figure 6, the gold reward (or even the oracle reward) does not drop for most cases even with the maximum KL distance considered. If a larger N is considered for BoN sampling, will the graph drop down as in Gao et al [2]? For a larger N is RewardMATH still successful in detecting more robust reward models regarding the overoptimization problem?

---

> ### Author Response · Authors · 2024-11-17
> **Response to Reviewer 1Mfa [1/3]**
>
> Dear Reviewer 1Mfa,
>
> Thank you for your deep understanding of our work, and appreciate your suggestions that can make our work stronger. We will address the key concern raised in the review below.
>
> ___
> ### **W1. Expansion to additional domains**
>
> We sincerely thank you for your suggestion, which can further enhance our work. However, it seems there are some misunderstandings about what we intended to convey in our work. If our main goal were to propose a well-crafted new benchmark, focusing solely on mathematics might limit the scope of the research; however, our goal is to provide insights into future directions for constructing reliable benchmarks for reward models. And it is fairly intuitive that a reliable benchmark should not be vulnerable to reward hacking and that conducting multiple comparisons, rather than one-to-one-comparisons, provides a more reliable evaluation of reward models. So, we trust that it is important to thoroughly validate our design through at least one specific domain that allows for in-depth experiments and analysis. We outline the reason behind choosing the mathematics domain for this work below:
>
> * **One of the tasks where reward models are most extensively used is mathematics reasoning.** Since the success of RLHF, many studies have utilized reward models extensively. In particular, mathematical reasoning tasks have increasingly employed reward models both during training to enhance reasoning capabilities and during inference using reward-based techniques such as Best-of-N (BoN) sampling or Monte Carlo Tree Search (MCTS). This justifies the analysis of reward models within the context of mathematical reasoning tasks.
> * **Mathematical reasoning includes a clear human preference.** By selecting the math domain, where human preferences can be relatively easily defined by correctness, we were able to focus more effectively on the analysis.

---

> ### Author Response · Authors · 2024-11-17
> **Response to Reviewer 1Mfa [2/3]**
>
> ___
> ### **W2. Concerns about the lack of PPO Experiments and its impact on benchmark validation**
>
> Thank you for the insightful comments and for recognizing the extensive experiments conducted using Best-of-N (BoN) sampling. We understand the concern regarding the lack of PPO or DPO in our experiment setup and appreciate the opportunity to address this. Below, we would like to address any concerns and clarify our experimental setup:
>
> 1. **Details of synthetic setup for PPO:**
>    In response to your question regarding the synthetic setup in Figure 5, we provide further details here. We trained on a 12K MATH dataset for 2000 steps and saved a total of 10 checkpoints at 200-step intervals. For each checkpoint, we computed the KL divergence, oracle reward, and gold reward. Following \[1\], the fitted dotted curves utilize scaling laws proposed in \[2\], following the formula $R\_{RL}(d) \= d(\\alpha\_{RL}-\\beta\_{RL}\\log\_{d})$.
>    Through the fitted dotted curve, we observed reward overoptimization relative to data scale, a phenomenon experimentally demonstrated in many studies. From the results of this experiment (Figure 5 and Table 3), we found that RewardBench does not reflect the degree of overoptimization.
> 2. **Challenges of PPO in a non-synthetic setup:**
>    Many previous studies have used the responses of an SFT model to train the same pretrained model as the reward model to achieve stable RLHF (PPO) training \[3-4\]. In particular, \[4\] highlights that initializing the reward model with the same pretrained model helps prevent information mismatches with the policy model, contributing to a consistent and accurate reward signal. Additionally, \[4\] and \[5\] suggest that as the policy model improves, the data distribution shifts, and if the reward model is not exposed to this new distribution, its accuracy may be limited.
>    In our case, the reward models we evaluate are trained on different backbone (i.e. pretrained) models and are also different from the policy model, making stable PPO training challenging in a non-synthetic setup. Indeed, when we attempted training with several reward models, the training process was highly unstable. Due to these reasons, it was difficult to perform comprehensive PPO experiments with various reward models.
>    We have now clarified the challenges of PPO in a non-synthetic setup in Appendix B.4 of the updated PDF **([click to see the pdf \[link\]](https://openreview.net/pdf?id=0er6aOyXUD))**.
> 3. **Correlation with downstream tasks other than BoN:**
>    We understand the reviewer’s concern regarding the need to verify the effectiveness of RewardMATH beyond BoN sampling, through methods such as DPO or PPO. As previously mentioned, due to the instability of PPO experiments in our setup, we focused on conducting experiments where the reward model can effectively provide learning signals.
>    * **Preference data for DPO constructed using the reward model:** We created a preference dataset for DPO by selecting a response with the highest reward as the “chosen” sample and a response with the lowest reward as the “rejected” sample.
>
>    In our experiments, we used MetaMATH-Mistral-7B as the SFT model and selected a 32K subset of data from the MetaMATH dataset as the training dataset, considering the short discussion (rebuttal) period. We performed n=32 sampling with the SFT model and removed instances that were entirely correct or incorrect to reduce noise and better assess whether the reward model provides meaningful learning signals. Finally, we obtained rewards from each reward model for a final dataset of 13.5K responses and conducted training with DPO.
>
>    The table below presents the correlation between the results of the optimized policies on MATH500 and the benchmark results. As a result, we reconfirm that the results on DPO also show a stronger correlation than RewardBench.
>
> |  | RewardBench | RewardMATH |
> | :---- | ----- | ----- |
> | DPO with reward model | 0.156 | 0.725 |
> | BoN | 0.187 | 0.902 |
>
> **References**
>
> [1] Rafailov, Rafael, et al. "Scaling laws for reward model overoptimization in direct alignment algorithms." arXiv preprint arXiv:2406.02900 (2024).
>
> [2] Gao, Leo, John Schulman, and Jacob Hilton. "Scaling laws for reward model overoptimization." Proceedings of the 40th International Conference on Machine Learning. 2023.
>
> [3] Ouyang, Long, et al. "Training language models to follow instructions with human feedback." Advances in neural information processing systems 35 (2022): 27730-27744.
>
> [4] Touvron, Hugo, et al. "Llama 2: Open foundation and fine-tuned chat models." arXiv preprint arXiv:2307.09288 (2023).
>
> [5] LeVine, Will, et al. "A Baseline Analysis of Reward Models' Ability To Accurately Analyze Foundation Models Under Distribution Shift." arXiv preprint arXiv:2311.14743 (2023).

---

> ### Author Response · Authors · 2024-11-17
> **Response to Reviewer 1Mfa [3/3]**
>
> ___
> ### **W3. The agreement between RewardMATH or RewardBench and oracle reward scores in Figure 6.**
>
> Thank you for your constructive suggestion regarding the statistical analysis for measuring agreement. Typically, reward overoptimization is illustrated with a graph, as shown in Figure 6. However, unlike previous studies, we examined a wide range of reward models, which may obscure clear trends. Therefore, we agree that a statistical analysis, such as Kendall's tau, would be beneficial for providing deeper insights. We calculate Kendall’s tau for the results in Figure 6, comparing the reward scores (gold and oracle reward) with the benchmark performance (RewardBench and RewardMATH) at specific KL divergence (i.e. specific $n$).
>
> | $n$ | KL | RewardBench (gold) | RewardMATH (gold) | RewardBench (oracle) | RewardMATH (oracle) |
> | :---- | :---- | :---- | :---- | :---- | :---- |
> | 64 | 3.17 | 0.400 | 0.718 | 0.116 | 0.692 |
> | 128 | 3.86 | 0.348 | 0.718 | 0.182 | 0.761 |
> | 256 | 4.55 | 0.322 | 0.692 | 0.156 | 0.761 |
>
> ___
>
> ### **Q1.**
> > As the proxy reward model trained from synthetic data shows far from optimal performance in Table 3 (only around 13% for RewardMATH and 69% for RewardBench), can you consider using better proxy RMs? The increase in value from 12.68 to 13.51 is not very convincing to me that this is a strong trend.
>
> We agree that the proxy reward model’s performance, as shown in Table 3, is not optimal, particularly with results of approximately 13% for RewardMATH and 69% for RewardBench. However, we believe this is not critical, as our experiment follows the original experimental setups in \[6\] and \[7\]. The primary goal of this experiment is to observe reward overoptimization as data size increases in a synthetic setup and to verify whether this trend is well-reflected by the score on a benchmark designed for reward models. Therefore, using a more advanced proxy RM was not considered in this work. Testing with a stronger proxy RM would require a broader dataset and diverse training approaches, which would fall outside of a synthetic setup.
>
> **References**
>
> \[6\] Gao, Leo, John Schulman, and Jacob Hilton. "Scaling laws for reward model overoptimization." Proceedings of the 40th International Conference on Machine Learning. 2023\.
>
> \[7\] Coste, Thomas, et al. "Reward Model Ensembles Help Mitigate Overoptimization." The Twelfth International Conference on Learning Representations.
>
> ___
> ### **Q2.**
> > If a larger N is considered for BoN sampling, will the graph drop down as in Gao et al \[2\]? For a larger N, is RewardMATH still successful in detecting more robust reward models regarding the overoptimization problem?
>
> We appreciate your question regarding the impact of larger N on the trend in Figure 6\. Due to constraints on available computational resources, we evaluate BoN sampling using only $n=256$. As N continues to increase, a robust reward model is likely to either converge or exhibit minimal overopimization at higher KL divergence. However, since reward models that overoptimize at low KL divergence or have lower peaks tend to perform poorly on RewardMATH, we expect that RewardMATH will remain effective at detecting such models even with large N.

---

> ### Comment · Reviewer_1Mfa · 2024-11-21
>
> Thank you for your response.
>
> [W1] I understand that this work does not feature a new benchmark and it proposes a new method that researchers may use when creating benchmarks. However, I think this raises the natural question: will this approach (multi-to-one comparisons) generalize to other domains than Math, on more complex domains such as coding or safety or helpfulness? I understand that for validation, a clear human preference is needed, which I think is definitely possible for simple coding domains (utilize unit tests provided by HumanEvalPack [1]), and likely possible for safety (one could use Llama-Guard [2]). If this work indeed proposes a new method, I think it should be verified on at least another domain. The math dataset used seems too simple for me to truly acknowledge this method's capability.
>
> I understand that this may not be possible within the rebuttal period due to time constraints. But I believe this remains a weakness.
>
> ---
>
> [W2, W3]
> Thank you for providing additional experiments and the statistical analysis. I think (although limited compared to BoN), the results do confirm that this method does work on other RLHF methods. Were there any differences observed between the results of BoN and DPO?
>
> ---
>
> [Q2] My question was regarding whether the graph will drop down after the peak. For example, in Figure (6) (a) of your original manuscript, the Intern LM-2-7B and other RMs show a rather linear graph whereas for Gao et al [3], the curve is a flipped U-shape, even for most competent RMs. My question is whether there is a reason for this seemingly different behavior of the two curves. For example, in the DPO experiment you have conducted, the fine-tuned LLM will likely have a larger KL divergence than BoN sampling. Have you drawn this curve using your version of DPO? And does the shape follow that of Gao et al? If there is a difference, why so?
>
> [1] https://huggingface.co/datasets/bigcode/humanevalpack
>
> [2] https://ai.meta.com/research/publications/llama-guard-llm-based-input-output-safeguard-for-human-ai-conversations/
>
> [3] Gao, Leo, John Schulman, and Jacob Hilton. Scaling laws for reward model overoptimization. Proceedings of the 40th International Conference on Machine Learning. 2023.

---

> ### Author Response · Authors · 2024-11-21
> **Official Comment by Authors**
>
> Thank you for your response!
>
> ### **[W1]**
>
> We agree with your suggestion that conducting analysis across diverse domains would have made our work more comprehensive and engaging. However, we believe that the following findings are applicable beyond the specific domain of mathematics:
>
> * In various domains beyond mathematics, it is important that benchmarks for reward models are validated both **based on** **their correlation with the performance on downstream tasks** and **through the lens of reward overoptimization.**
> * While we demonstrated **the effectiveness of one-to-many comparisons over one-to-one comparisons in assessing the robustness of reward models** in mathematics, accurately ranking multiple responses, rather than merely comparing two, is a design that can also be applied to other domains.
>
> While we did not explore multiple domains, these findings can be considered applicable to other domains. Thank you once again for your constructive suggestion for improvement.
>
> &nbsp;
>
> **[Updates]** We provide additional clarification on **the key insights** of this work in the general response.
>
> ___
>
> ### **[W2, W3]**
>
> The primary distinction lies in the reward signals used for optimization: BoN focuses solely on selecting the response with the highest reward, while DPO leverages both the highest-reward and lowest-reward responses during training. Consequently, a reward model that assigns high rewards to correct solutions performs well in BoN, whereas one that also avoids assigning low rewards to correct solutions stands out in DPO.
>
> Given these differences, achieving a strong correlation for DPO through one-to-many comparisons is challenging. We believe that adopting many-to-many comparisons, as outlined in the future work (Appendix A), will better capture these and lead to stronger correlations.
>
> ___
> ### **[Q2]**
>
> Thank you for your question. The differences in graph shapes stem from variations in the experimental design. In [1], a gold reward model with a 6B parameter size was used, while the proxy reward model was of the same architecture but with a different size. As observed in Figure 1 of [1], a very small model (3M) produces a flipped U-shape curve, whereas a larger model, closer in size to the gold reward model (3B), shows a nearly linear graph. This suggests that when the performance of the proxy reward model closely matches that of the gold reward model, the graph of the results tends to appear linear. Similarly, in Figure 6 of our paper, the reward models analyzed do not exhibit significant differences in size or structure, which leads to a more linear graph.
>
> However, for oracle rewards, where the performance is poor and reward overoptimization occurs rapidly, we observe reward collapse in certain models. The lack of a flipped U-shape in such cases is also due to experimental design differences. In [1], extensive resources were used to conduct experiments with $N=60,000$, resulting in a smooth curve with densely sampled points. In contrast, our experiments were conducted with fewer $N$, providing discrete points that were connected linearly, which likely contributes to the differences you mentioned. Notably, Figure 2 of [2] and Figure 7 of [3] present similar discrepancies, suggesting that such discrepancies are common in related studies.
>
> For the DPO experiments, KL divergences are calculated based on specific checkpoints. To achieve the curve seen in [1], KL needs to be sampled at very fine intervals, which is challenging to obtain from the checkpoints. A similar pattern is also observed in Figure 1 of [4].
>
> **References**
>
> \[1\] Gao, Leo, John Schulman, and Jacob Hilton. "Scaling laws for reward model overoptimization." Proceedings of the 40th International Conference on Machine Learning. 2023\.
> \[2\] Yang, Rui, et al. "Regularizing Hidden States Enables Learning Generalizable Reward Model for LLMs." arXiv preprint arXiv:2406.10216 (2024).
> \[3\] Rame, Alexandre, et al. "WARM: On the Benefits of Weight Averaged Reward Models." Forty-first International Conference on Machine Learning.
> \[4\] Rafailov, Rafael, et al. "Scaling laws for reward model overoptimization in direct alignment algorithms." arXiv preprint arXiv:2406.02900 (2024).

---

> ### Author Response · Authors · 2024-11-28
> **Sincere Request for Review of Our Responses and Revised Paper**
>
> Dear Reviewer 1Mfa,
>
> Thank you once again for your time and effort in providing insightful feedback on our paper. As you highlighted, we have addressed the expandability to other domains by including additional details in Appendix A of the updated PDF **([click to see the pdf \[link\]](https://openreview.net/pdf?id=0er6aOyXUD))** . Furthermore, we have provided responses to your additional questions.
>
> We hope that our responses sufficiently address your concerns. Should there be an opportunity for further discussion during the rebuttal period, we would be delighted to engage and provide any clarifications or further insights.
>
> Best regards,
>
> The Authors of Paper 10281

---

> ### Author Response · Authors · 2024-12-02
> **Gentle Reminder**
>
> Dear Reviewer 1Mfa,
>
> Thank you again for your thoughtful feedback on our paper. Your feedback has been invaluable in improving our work. As today is the final day for discussion, we would be delighted to provide any clarifications or further insights if needed. Please let us know if there are any remaining concerns we can address.
>
> Best regards,
>
> The Authors of Paper 10281

---

### Official Review · Reviewer_bxC4 · 2024-11-07

**Soundness:** 2
**Presentation:** 2
**Contribution:** 2
**Rating:** 6
**Confidence:** 4

**Summary:**

Authors aims to design a better benchmark for evaluating reward models in reasoning tasks. Authors identify problems with the previous benchmark RewardBench and proposes RewardMath:
* RewardBench is based on PRM800K which contains many wrong annotations, RewardMath instead is based on MATH500.
* RewardBench uses pairs of (human annotated example, LLM annotated incorrect example).  RewardMath includes more than one incorrect example.
* RewardBench’s accepted and rejected examples have different number of steps, this could be a spurious correlation that leads to reward hacking. RewardMath fixes this.
* RewardBench’s PRM evals uses product instead of mean which biases shorter responses, authors fix this by using mean instead.

To demonstrate the improvements of rewardmath 1) authors compare performances of different reward models on both RewardBench and RewardMath, and show rewardmath has higher correlation with downstream evals 2) authors show rewardmath exhibits the common trend of larger dataset -> better performance while rewardbench does not.

**Strengths:**

* The topic on how to Improve LLM reasoning capabilities has recently gained a lot of attention. This paper focuses on having good benchmarks for evaluating these efforts, and this could be very impactful if done correctly.

* Authors identify flaws of existing benchmarks and make good efforts to fix them.

* Paper has good results, specifically Figure 4 is very cool showing RewardMath has stronger correlation with downstream tasks.

**Weaknesses:**

See questions I have below

**Questions:**

* RewardMath is based on the dataset MATH500, where does the dataset MATH500 come from? Is MATH500 prior work (and if yes the citation is missing) or is this a contribution of the paper (in this case it should be made clear).
* Does MATH500 address the incorrect annotation problem found in PRM800K?
* Can authors also show evaluations and ablations on gsm8k [1] and MATH [2] which are the most common eval tasks for LLM reasoning capabilities?
*  Authors identify that in RewardBench the accepted response often has less steps than rejected ones, which could give a chance for models to reward hack (i.e. reward relies on the number of steps instead of the actual response quality). Did the authors ablate this? I.e.  Does the reward-hacking model predict lower reward if we make the accepted response in RewardMath longer? And vice versa, does it predict higher reward if we make the rejected response shorter?
* Regarding RewardMath giving more than one rejected responses: If one is trying to do preference learning using a llama model as the base model, is it important for the reward model to know the rejected response generated by a non-llama model should be worse than the accepted response? I.e. the distribution could be very different that it never encounters it during preference learning. I.e. for Figure 4, if we use a llama model as the policy, does one-to-many RewardMath still do better than one-to-one RewardMath chosen & Llama rejection?
* Does reward-model free alignment methods like DPO also suffer from reward model overfitting problem?  What is the advantage of using reward models over reward model-free methods for reasoning tasks?
* Does the benchmark evaluate cases where both the rejected and chosen response arrive at the same answer, but the rejected answer has the wrong steps? I.e. this is common for truth or false questions.


[1] Training Verifiers to Solve Math Word Problems

[2]  Measuring mathematical problem solving with the math dataset

---

> ### Author Response · Authors · 2024-11-17
> **Response to Reviewer bxC4 [1/3]**
>
> Dear Review bxC4,
>
> We appreciate your thoughtful review and recognition of our efforts to advance benchmarks for reward models. We hope that the following responses help clarify any ambiguities in the paper and make our work more comprehensive:
> ___
> ### **Q1-2. About MATH500 dataset**
> We apologize for any confusion caused by the lack of detailed explanation regarding MATH500 in our initial submission. We have now included citation and comprehensive information in Section 3 and Appendix B of the updated PDF **([click to see the pdf \[link\]](https://openreview.net/pdf?id=0er6aOyXUD))**.
>
> MATH500 is derived from the original MATH dataset [1], which comprises a 7.5K training set and a 5K test set. During the development of PRM800K [2], the initial 7.5K training set was insufficient for training a robust Process Reward Model (PRM) on step-by-step solution data. Consequently, 4.5K problems from the MATH test set were incorporated into the training set, leaving a remaining subset of 500 problems now referred to as MATH500. Since the release of PRM800K, MATH500 has been widely adopted to prevent overlap between training and test sets.
> To clarify, MATH500 is part of the original MATH dataset and is not a new contribution of our paper. Moreover, only PRM800K contains mis-annotated samples, while the MATH500 derived from the original MATH dataset remains unaffected.
>
> **References**
>
> [1] Hendrycks, Dan, et al. "Measuring Mathematical Problem Solving With the MATH Dataset." Thirty-fifth Conference on Neural Information Processing Systems Datasets and Benchmarks Track
>
> [2] Lightman, Hunter, et al. "Let's Verify Step by Step." The Twelfth International Conference on Learning Representations.
> ___
>
> ### **Q3. Evaluations and ablations on GSM8K and MATH**
> As previously mentioned, MATH500 is a subset of the MATH dataset, and our experiments already include evaluations on this dataset. While we appreciate the suggestion to include evaluations on GSM8K, we chose to exclude it from our current study for the following reasons:
> Simplicity and Established Performance:  GSM8K is a relatively simple dataset on which many studies have already achieved high scores, potentially limiting the meaningful insights gained from further evaluation.
> Data Distribution Concerns: In our experiments, we classify mathematical reasoning test sets based on the data learned by the RM as either in-distribution (e.g., MATH500) or out-of-distribution (e.g., Gaokao-math). Given GSM8K's widespread use and its potential inclusion in training data, it is challenging to classify it definitively as either in-distribution or out-of-distribution.
> ___
>
> ### **Q4. An ablation study on the score based on the number of steps.**
> Thank you for your insightful question. Since both RewardBench and RewardMATH are based on MATH500, with RewardBench using human-annotated chosen solutions and RewardMATH using machine-generated ones, we ablate the scores of chosen solutions across each benchmark. The first table below shows the average number of steps for chosen solutions in each benchmark, with RewardBench—which comprises human chosen solutions—having a lower average step count. The second table below presents the average scores of chosen solutions generated by reward models, revealing that many models tend to assign higher scores to shorter-step answers and indicating that training was aimed at achieving high scores within RewardBench.
> | RewardBench mean length | RewardMATH mean length |
> | ----- | ----- |
> | 4.11 | 5.99 |
>
> |  Model | Mean score of chosen solution |  |
> | ----- | ----- | ----- |
> |  | **RewardBench** | **RewardMATH** |
> | ArmoRM-Llama3-8B-v0.1 | 8.21 | 8.04 |
> | Skywork-Reward-Llama3.1-8B | 202.22 | 47.13 |
> | Oasst-rm-2.1-pythia-1.4b | 17.97 | \-25.65 |
> | Internlm2-20b-reward | 25.78 | 10.75 |
> | Internlm2-7b-reward | 61.17 | 39.81 |
> | GRM-llama3-8B | \-10.77 | 86.50 |
> | GRM-gemma-2B | 4.61 | 45.80 |
> | Eurus-RM-7b | \-5908.98 | \-5172.88 |
> | Beaver-7b-v2.0-reward | 26.62 | 20.14 |

---

> > ### Comment · Reviewer_bxC4 · 2024-11-21
> > **Thank you Authors**
> >
> > Dear Authors,
> >
> > Thank you for your detailed rebuttal and your detailed explanations towards my questions and concerns.
> >
> > Some remaining concerns I have regarding evaluations:
> >
> > > As previously mentioned, MATH500 is a subset of the MATH dataset, and our experiments already include evaluations on this dataset.
> >
> > Why not just use the full MATH dataset since this is the standard practice, instead of just the MATH500 subset?
> >
> > >GSM8K is a relatively simple dataset on which many studies have already achieved high scores, potentially limiting the meaningful insights gained from further evaluation.
> >
> > I see that classifier-based reward models models (in Table 2) as well as the base models (in Figure 4) used in the paper are mostly  on the scale of under 8B parameters. On this scale, the models usually are not "achieving high scores" on gsm8k (i.e. <80%). So I still think it's valuable to include those results.
> >
> > > Given GSM8K's widespread use and its potential inclusion in training data, it is challenging to classify it definitively as either in-distribution or out-of-distribution.
> >
> > I don't know if it's *that* important whether we can give a hard classification of it's "in" or "out of distribution". Rather I think it would be more important to include both of these evals because they are the most commonly used benchmark for math-related tasks.

---

> ### Author Response · Authors · 2024-11-17
> **Response to Reviewer bxC4 [2/3]**
>
> ___
> ### **Q5.**
> > If one is trying to do preference learning using a llama model as the base model, is it important for the reward model to know the rejected response generated by a non-llama model should be worse than the accepted response?
>
> Thank you for your insightful question. If we only consider reward models for preference learning, it may not be important for the reward model to recognize rejected responses from non-Llama models (i.e. different base models). However, creating a benchmark based solely on responses from one model could limit its applicability and generalizability across different models and scenarios. In this work, we collected responses from various models to ensure that the reward model can be applied across multiple scenarios (e.g. inference-time optimization with the reward model, PPO, and dataset construction with the reward model),  allowing us to evaluate its performance. In this context, we also conducted an analysis of LLM-as-a-judge.
> In our experiments (Section 4), we used MetaMATH-Mistral-7B and WizardMATH-7B-v1.1 as policy models and observed a correlation between the results of rejected responses from these models (the 4th and 5th rows from the bottom in Table 12\) and the Best-of-N (BoN) results. Additionally, Table 11 (Appendix C.4) presents the correlations between the performance of optimized policy on the downstream tasks and the performance of reward models on a dataset where the policy’s solutions were removed from RewardMATH. From these results, we observe that one-to-one comparisons can be significantly influenced by the policy models used in the experiments, as noted by the reviewer. In contrast, one-to-many comparisons remain unaffected by this influence and exhibit a strong correlation.
>
> ___
> ### **Q6.**
> > Do reward model-free alignment methods like DPO experience overfitting issues, and what advantages do reward models offer over such methods for reasoning tasks?
>
> Thank you for pointing this fundamental question out, as it addresses key considerations in understanding the limitations of reward model-free alignment methods like DPO and the potential advantages of using reward models, especially for complex reasoning tasks. \[3\] demonstrates that reward overoptimization can occur even in direct alignment algorithms (e.g. DPO).
> Additionally, using reward models is advantageous over model-free alignment methods for the following reasons:
>
> * **PPO improves over DPO for reasoning:** \[4\] empirically shows that PPO achieves a larger performance improvement over DPO in reasoning tasks. And \[5\] investigates the limitations of DPO through theoretical and experimental analysis, finding that DPO is sensitive to distribution shifts between base model outputs and preference data, highlighting a fundamental limitation of DPO. The proposed PPO method in \[5\] shows performance improvements in reasoning tasks, and the paper also notes the critical role of the reward model during training.
> * **Using reward models for inference-time scaling:** \[6\] shows the importance of inference-time scaling, demonstrating that applying inference-time scaling with an oracle verifier yields very high performance. \[7\] and \[8\] also demonstrate performance improvements by using reward models in different inference-time methods.
>
> **References**
>
> \[3\] Rafailov, Rafael, et al. "Scaling laws for reward model overoptimization in direct alignment algorithms." *arXiv preprint arXiv:2406.02900* (2024).
>
> \[4\] Ivison, Hamish, et al. "Unpacking DPO and PPO: Disentangling Best Practices for Learning from Preference Feedback." arXiv preprint arXiv:2406.09279 (2024).
>
> \[5\] Xu, Shusheng, et al. "Is DPO Superior to PPO for LLM Alignment? A Comprehensive Study." Forty-first International Conference on Machine Learning.
>
> \[6\] Brown, Bradley, et al. "Large language monkeys: Scaling inference compute with repeated sampling." arXiv preprint arXiv:2407.21787 (2024).
>
> \[7\] Kang, Jikun, et al. "Mindstar: Enhancing math reasoning in pre-trained llms at inference time." arXiv preprint arXiv:2405.16265 (2024).
>
> \[8\] Wu, Yangzhen, et al. "An Empirical Analysis of Compute-Optimal Inference for Problem-Solving with Language Models." arXiv preprint arXiv:2408.00724 (2024).

---

> ### Author Response · Authors · 2024-11-17
> **Response to Reviewer bxC4 [3/3]**
>
> ___
> ### **Q7.**
> > Does the benchmark evaluate cases where both the rejected and chosen response arrive at the same answer, but the rejected answer has the wrong steps?
>
> No, we initially considered including rejected solutions with correct answers but wrong reasoning steps; however, we decided not to include them for the following reasons:
>
> * **Challenges in Data Collection:** Our primary focus was to analyze the design of reliable benchmarks for reward models. Therefore, when collecting data, we filtered based on the correctness of the final answer, followed by a manual inspection conducted by humans (authors) to ensure data quality. If we were to include such cases, however, this process would become more costly.
> * **Rarity of Such Cases:** In the MATH dataset, True/False questions are rare, as are cases where the answer is correct but the reasoning is incorrect. For instance, in the problem where the task is to count the possible values of $n$, the correct solution may involve $n=4,5,6$, resulting in a final answer of 3\. However, a model generated solution consisting of $n=2,3,4$ could still arrive at the same final answer of 3\.

---

> ### Author Response · Authors · 2024-11-21
> **Sincere Request for Review of Our Responses, New Experiments, and Revised Paper**
>
> Dear Reviewer bxC4,
>
> Thank you again for your time and effort to provide your insightful feedback on our paper.
>
> We have addressed your comments and added comprehensive details about MATH500 in Section 3 and Appendix B of the updated draft **([click to see the pdf \[link\]](https://openreview.net/pdf?id=0er6aOyXUD))**.
> If you have any remaining questions or require further clarification, we would be happy to address them before the time window closes.
>
> Thank you so much for your time and valuable feedback!
>
> Best regards,
>
> The Authors of Paper 10281

---

> ### Author Response · Authors · 2024-11-23
>
> Thank you for your response!
>
> > Why not just use the full MATH dataset since this is the standard practice, instead of just the MATH500 subset?
>
> As previously mentioned, many studies aiming to improve mathematical reasoning capabilities through reward models primarily use MATH500 instead of the full MATH test set to prevent training set overlap with PRM800K [1-4]. Since PRM800K incorporates parts of the MATH test set into its training set, using MATH as a test set could compromise evaluation reliability. To address this, MATH500 [1], designed to exclude such overlap, has become the standard for testing. Even the OpenAI-O1 report includes the performance of MATH500 [5]. Finally, [2] said that “The subset consists of 500 representative problems, and we find that the subset evaluation produces similar results to the full-set evaluation”.
>
> Therefore, MATH500 serves as a reliable and unbiased test set for evaluating mathematical reasoning capabilities within the community.
>
> **References**
> \[1\] Lightman, Hunter, et al. "Let's Verify Step by Step." The Twelfth International Conference on Learning Representations.
> \[2\] Wang, Peiyi, et al. "Math-shepherd: Verify and reinforce llms step-by-step without human annotations." Proceedings of the 62nd Annual Meeting of the Association for Computational Linguistics (Volume 1: Long Papers). 2024\.
> \[3\] Sun, Zhiqing, et al. "Easy-to-hard generalization: Scalable alignment beyond human supervision." arXiv preprint arXiv:2403.09472 (2024).
> \[4\] Setlur, Amrith, et al. "Rewarding Progress: Scaling Automated Process Verifiers for LLM Reasoning." arXiv preprint arXiv:2410.08146 (2024).
> \[5\] [https://openai.com/index/learning-to-reason-with-llms/](https://openai.com/index/learning-to-reason-with-llms/)
> ___
> > I see that classifier-based reward models models (in Table 2) as well as the base models (in Figure 4) used in the paper are mostly on the scale of under 8B parameters. On this scale, the models usually are not "achieving high scores" on gsm8k (i.e. <80%). So I still think it's valuable to include those results.
>
> > I don't know if it's that important whether we can give a hard classification of it's "in" or "out of distribution". Rather I think it would be more important to include both of these evals because they are the most commonly used benchmark for math-related tasks.
>
> We appreciate your feedback and understand your suggestions that including the results on GSM8K would enhance this work. Accordingly, we also examine the results on the GSM8K dataset as you commented. The first table below exhibits the performance of Mistral-MetaMATH-7B, used as the policy model in Figure 4, on GSM8K. Moreover, another policy model we used, WizardMath-7B-v1.1, has already achieved a score of 83.2 (pass@1), and Llama-3.1-8B-Instruct, though not a math-specialized model, has reached 84.5 (pass@1) on GSM8K.
>
> | N | pass@N |
> | :---: | :---: |
> | 1 | 78.772 |
> | 2 | 79.909 |
> | 4 | 82.942 |
> | 8 | 85.747 |
> | 16 | 88.779 |
> | 32 | 91.812 |
> | 64 | 94.011 |
> | 128 | 96.361 |
> | 256 | 98.939 |
>
> However, as you pointed out, the results on GSM8K would be invaluable; therefore, we further conducted an ablation study on it. The second table below presents the correlation between the results of BoN sampling on GSM8K and the performance on each benchmark. Due to limited GPU resources, the ReasonEval-34B experiment is still in progress and will be included in the updated PDF once it is completed. To ensure a fair comparison, we have also provided correlations for the MATH dataset without ReasonEval-34B. The results demonstrate that our design also presents a strong correlation for GSM8K, while RewardBench has a weak correlation.
>
> |  | RewardBench | RewardMATH |
> | :---- | ----- | ----- |
> | GSM8K (BoN, w/o ReasonEval-34B) | 0.221 | 0.818 |
> | MATH (BoN, w/o ReasonEval-34B) | 0.368 | 0.946 |
> | MATH (BoN) | 0.187 | 0.902 |

---

> ### Author Response · Authors · 2024-11-28
> **Sincere Request for Review of Our Responses and Revised Paper**
>
> Dear Reviewer bxC4,
>
> Thank you once again for your time and effort to provide insightful feedback on our paper. As you pointed out, the results on GSM8K would be invaluable and we are pleased to inform you that the ReasonEval-34B experiment has now been completed, and the results have been incorporated into the updated PDF **([click to see the pdf \[link\]](https://openreview.net/pdf?id=0er6aOyXUD))**. The table below presents the correlation between the BoN sampling results on GSM8K and the performance on each benchmark. As demonstrated, our design continues to show a strong correlation for GSM8K, whereas RewardBench exhibits a weaker correlation.
>
> |  | RewardBench | RewardMATH |
> | :---- | ----- | ----- |
> | GSM8K (BoN) | 0.209 | 0.797 |
> | MATH (BoN) | 0.187 | 0.902 |
>
> We hope these additional results address your concerns. If there is an opportunity for further discussion within the rebuttal period, we would be happy to engage and provide any clarifications or further insights.
>
> Best regards,
>
> The Authors of Paper 10281

---

> ### Author Response · Authors · 2024-12-02
> **Gentle Reminder**
>
> Dear Reviewer bxC4,
>
> Thank you again for your thoughtful feedback on our paper. Your feedback has been invaluable in improving our work. As today is the final day for discussion, we would be delighted to provide any clarifications or further insights if needed. Please let us know if there are any remaining concerns we can address.
>
> Best regards,
>
> The Authors of Paper 10281

---

### Author Response · Authors · 2024-11-17
**General Response to All Reviewers**

We greatly appreciate all reviewers for their time in reviewing our paper and for providing thoughtful suggestions to make our paper stronger. We would like to address comments and suggestions commonly raised by reviewers.


___
## Clarification for Our Main Contributions
Through the reviewers’ comments, we realized that the paper placed too much emphasis on the new benchmark, RewardMATH. However, the purpose of RewardMATH—a simplified benchmark representing our proposed design—was to validate the reliability of our benchmark design in comparison to RewardBench, rather than to serve as a widely adopted benchmark in future studies. To clarify this point, we slightly revised Title, Abstract and Introduction to reduce the emphasis on RewardMATH, aligning more closely with our original intent.

The contributions of the paper are as follows:

- We identify the issues with existing benchmark for reward models (i.e. RewardBench)—such as poor quality, vulnerability to reward hacking, and the risk of misjudgement due to isolated cases—and introduce a new design for reliable benchmark for reward models, focusing on reducing the risks of reward hacking and employing multiple comparisons to effectively estimate the degree of reward overoptimization.
- We conduct extensive experiments validating that the scores on RewardMATH strongly correlates with the performance of optimized policy and effectively estimates the degree of reward overoptimization. These results pave the way for future directions in evaluating reward models more reliably.


Furthermore, **_our key insights_** can be described as follows:

- **In benchmarks for reward models, a significant difference between the chosen and rejected responses shows low correlation with downstream tasks due to the potential for reward hacking.**
- **One-to-one comparisons may yield inaccurate results depending on the preference pairs, which in turn results in low correlation with downstream tasks.**
- **A benchmark employing multiple comparisons can effectively capture reward overoptimization, indicating its ability to assess the robustness of reward models.**

___
## A Scope Limited to the Mathematical Reasoning
We believe this concern stems from a misunderstanding of our contribution. If our main goal were to propose a well-crafted new benchmark, focusing solely on mathematics might limit the scope of the research; however, our goal is to provide insights into future directions for constructing reliable benchmarks for reward models. And it is fairly intuitive that a reliable benchmark should not be vulnerable to reward hacking and that conducting multiple comparisons, rather than one-to-one-comparisons, provides a more reliable evaluation of reward models. So, we trust that it is important to thoroughly validate our design through at least one specific domain that allows for in-depth experiments and analysis.

Here, the reasons for choosing mathematical reasoning are summarized.

- **One of the tasks where reward models are most extensively used is mathematics reasoning.** In mathematical reasoning tasks, reward models are widely utilized during training to enhance reasoning capabilities and during inference-time using reward-based techniques such as best-of-$n$ (BoN) sampling or Monte Carlo Tree Search (MCTS).
- **Mathematical reasoning includes a clear human preference.** In mathematical reasoning, human preference can be easily defined as correctness, allowing us to focus effectively on the analysis without the need to deliberate over true preferences.

___
## Updates in the Revised Draft

The updated draft (click to see the pdf) also includes the following enhancements:

* We have updated the Title, Abstract and Introduction to clarify our main contributions.
* A detailed explanation of MATH500 has been added to line 200 and Appendix B.1.
* A comprehensive explanation of why PPO experiments were not conducted in a non-synthetic setup is discussed in Appendix B.4.
* The experiments of policy fine-tuning methods beyond BoN sampling have been featured in Appendix C.5.
* Additional experimental results based on factors considered in benchmark design have been integrated into Table 12, with analysis details added to Appendix C.4.

---

### Meta-Review · Area_Chair_bsE6 · 2024-12-21

**Metareview:**

**summary**

The paper introduces RewardMATH, a new benchmark for evaluating reward models in mathematical reasoning tasks, designed to address limitations in the existing RewardBench. RewardMATH uses a more accurate dataset (MATH500), multiple incorrect examples for robust evaluations, and a mean-based evaluation metric to avoid length bias, ensuring a fair comparison. It demonstrates improvements over RewardBench by showing higher correlation with downstream evaluations and validating the robustness of reward models through a one-to-many comparison approach. This approach enhances the reliability and effectiveness of reward models in identifying the most accurate responses in mathematical reasoning tasks.

**strengths**

* The authors successfully identify weaknesses in current benchmarks and introduce substantial improvements.
* The paper addresses the increasingly important topic of improving LLM reasoning capabilities by focusing on creating robust benchmarks for evaluating these efforts.
* Diverse reward models were evaluated on RewardMATH, demonstrating its versatility and wide applicability.

**weaknesses**

* The study's focus is narrowly tailored to mathematical problems, potentially overlooking other critical aspects of reward model. To validate the authors' claims about the generalizability of their method, applying the benchmark to other domains or more open-ended queries would be necessary, ideally moving beyond strictly mathematical reasoning.
* The paper's primary weakness lies in its limited experimental validation, relying mainly on correlation with best-of-n sampling. Broadening the scope to include other fine-tuning techniques like SFT, DPO, and PPO, and testing in other domains such as coding or open-ended generation tasks, could provide a more comprehensive assessment of the benchmarks' effectiveness across diverse applications.

**decision**

Although the authors emphasized that the main contribution of this paper extends beyond the new benchmark, it is crucial that the validation of key findings be further improved. I recommend that the authors address the concerns raised by reviewers and consider resubmitting to another venue after making necessary improvements.

**Additional Comments On Reviewer Discussion:**

The authors have successfully addressed several concerns raised by reviewers, including the incorporation of additional fine-tuning methods and evaluations on other math benchmarks, as well as addressing bias from specific LLM models. They have also revised the title, abstract, and main draft to better clarify the contributions of the paper. However, despite these improvements, several reviewers, including myself, remain skeptical about the overall soundness of the paper.

---

### Decision · Program_Chairs · 2025-01-22

Reject